# Sas-Ptp10D shapes germ-line stem cell niche by facilitating JNK-mediated apoptosis

**Kiichiro Taniguchi** *, **Tatsushi Igaki** *

Laboratory of Genetics, Graduate School of Biostudies, Kyoto University, Yoshida-Konoe-cho, Sakyoku, Kyoto, Japan

* taniguchi.kiichiro.3c@kyoto-u.ac.jp (KT); igaki.tatsushi.4s@kyoto-u.ac.jp (TI)

**Data Availability Statement:** All relevant data are within the manuscript and its Supporting Information files.

**Funding:** This work was supported by the JSPS KAKENHI (21K07120 and 16K07378 to KT), the

## Abstract

The function of the stem cell system is supported by a stereotypical shape of the niche structure. In *Drosophila* ovarian germarium, somatic cap cells form a dish-like niche structure that allows only two or three germ-line stem cells (GSCs) reside in the niche. Despite extensive studies on the mechanism of stem cell maintenance, the mechanisms of how the dish-like niche structure is shaped and how this structure contributes to the stem cell system have been elusive. Here, we show that a transmembrane protein Stranded at second (Sas) and its receptor Protein tyrosine phosphatase 10D (Ptp10D), effectors of axon guidance and cell competition via epidermal growth factor receptor (Egfr) inhibition, shape the dish-like niche structure by facilitating c-Jun N-terminal kinase (JNK)-mediated apoptosis. Loss of Sas or Ptp10D in gonadal apical cells, but not in GSCs or cap cells, during the pre-pupal stage results in abnormal shaping of the niche structure in the adult, which allows excessive, four to six GSCs reside in the niche. Mechanistically, loss of Sas-Ptp10D elevates Egfr signaling in the gonadal apical cells, thereby suppressing their naturally-occurring JNK-mediated apoptosis that is essential for the shaping of the dish-like niche structure by neighboring cap cells. Notably, the abnormal niche shape and resulting excessive GSCs lead to diminished egg production. Our data propose a concept that the stereotypical shaping of the niche structure optimizes the stem cell system, thereby maximizing the reproductive capacity.

## Author summary

Morphogenesis is a developmental process that reproducibly shapes complex structures from simple primordia. In this process, a variety of cellular processes including cytoskeletal changes, cell migration, cell division, and cell death cooperate to shape stereotypical structures. However, the mechanisms of how these processes contribute to build complex structures have been elusive. Here, by analyzing *Drosophila* ovarian development, a typical example of developmental shaping of a complex structure, we show that JNK-mediated apoptosis contributes to proper shaping of germ-line stem cell (GSC) niche. Mechanistically, transmembrane protein Sas and its ligand Ptp10D, effectors of cell competition, facilitate JNK-mediated apoptosis of developing gonadal apical cells that are adjacent to

JSPS KAKENHI (20H00515 and 21H05039 to TI), MEXT KAKENHI (21H05284 to TI), Japan Agency for Medical Research and Development (21gm5010001 to TI), AMED-CREST, Japan Agency for Medical Research and Development (22gm1710002h0001 to TI), the Takeda Science Foundation to TI, and the Naito Foundation to TI. The funders had no role in study design, data collection and analysis, decision to publish, or preparation of the manuscript.

**Competing interests:** The authors have declared that no competing interests exist.

GSC niche cells, thereby contributing to shaping stereotypical dish-like structure of the niche. Loss of apoptosis results in abnormal shaping of the GSC niche, which leads to excessive GSCs reside in the niche. Intriguingly, excessive GSCs results in diminished egg production, proposing a concept that the stereotypical shaping of the niche structure optimizes the function of the stem cell system.

## Introduction

Morphogenesis is a developmental process that reproducibly shapes complex structures from simple primordia. In this process, a variety of cellular processes including cytoskeletal changes, cell migration, cell division, and cell death cooperate to shape stereotypical structures essential for the organ functions [1–4]. A typical example of developmental shaping of a complex structure is the development of *Drosophila* ovarian germarium, where somatic cap cells act as major niche cells for germ-line stem cells (GSCs). In the process of germarium development, a cluster of cap cells shapes dish-like structure that restricts the number of residential GSCs to 2 or 3 [5–7]. Given that the direct anchoring of GSCs to the cap cell-cluster has been proposed to be important for GSC maintenance [8], the stereotypical shaping of the dish-like niche structure would be critical for appropriate operation of the stem cell system.

The development of the cap cell-cluster during *Drosophila* gonadogenesis begins at the late third-larval stage, where terminal filament cells form 10–15 rows in the middle of the gonad [9–11] (Fig 1A, left). At the pre-pupal stage, a part of apical cells neighboring terminal filament cells migrate into the space between each row of terminal filament and thereby physically separate them to individualize germaria (Fig 1A, center) [12]. At the same period, terminal filament cells at the base-side (germ cell-side) cause differentiation of neighboring interstitial cells into cap cells [7,10,13] (Fig 1A, center). Consequently, the apical cells spread-out as a cluster along the ovarian sheath to form the dish-like structure, which secrete a cytokine Decapentaplegic (Dpp) to maintain the stemness of GSCs [6,7] (Fig 1A, right). While the regulation of cap cell development is well understood, the mechanism by which the dish-like structure of the cap cell-cluster is shaped has been unclear.

Cell competition is a form of cell-cell interaction that causes context-dependent elimination of unfit viable cells through short-range cell-cell interaction [14–17]. For instance, oncogenic polarity-deficient cells such as *scribble* (*scrib*) or *discs large* (*dlg*) mutants overgrow by themselves but are eliminated from *Drosophila* epithelia when surrounded by wild-type cells, which is referred to as tumor-suppressive cell competition [18,19]. In this cell elimination process, an apical membrane protein Stranded at second (Sas) in wild-type cells interacts with its receptor Protein tyrosine phosphatase 10D (Ptp10D) in polarity-deficient cells, which causes inhibition of Epidermal growth factor receptor (Egfr) signaling in polarity-deficient cells to execute c-Jun N-terminal kinase (JNK)-dependent apoptosis [20]. Thus, Sas-Ptp10D signaling may act as a tumor-suppressor that promotes elimination of pre-malignant cells from epithelia. However, except for its role in axon guidance [21], the role of Sas-Ptp10D in the physiological contexts such as normal organ development has been elusive.

Here, we show that Ptp10D facilitates JNK-dependent apoptosis of gonadal apical cells via Egfr inhibition during gonadogenesis, which is essential for the proper shaping of the dish-like cap cell-cluster and thus the proper control of the GSC number. *Ptp10D* deficiency results in abnormal shaping of the niche structure, which leads to a significantly increased number of GSCs and diminished egg production. Our data proposes a physiological role of the cell competition machinery in tissue morphogenesis.

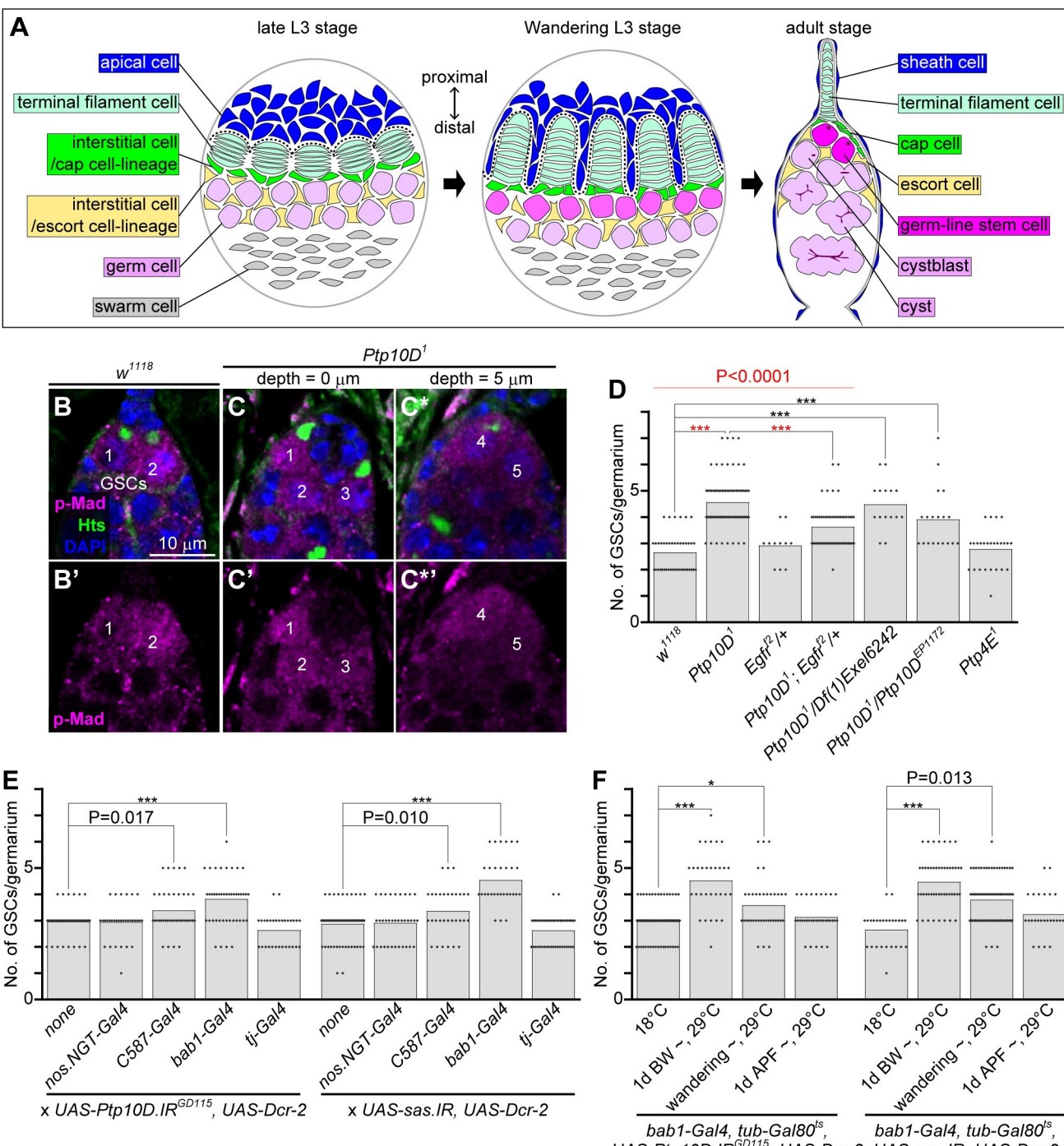

**Fig 1. Loss of *Ptp10D* or *sas* in gonadal apical cells causes excessive number of ovarian GSCs.** (A) A schematic diagram depicts the female gonadogenesis. Cell types constituting gonad are drawn by pseudo-colors as apical cells, terminal filament cells, cap cells/sheathe cells (interstitial cells), escort cells (interstitial cells), germ cells/germ-line stem cells/cystblasts/cysts, and swarm cells are colored with blue, pale green, green, light yellow, pink, and gray, respectively. Corresponding cell-types in larval and adult stages are drawn by same colors. (B and C) Proximal regions of female germaria 1 day after eclosion without mating labeled with anti-p-Mad antibody (magenta), anti-Hts antibody (green), and DAPI (blue). Proximal is to the top. *w*[1118] (B) and *Ptp10D*[1] homozygotes (C) with two and five GSCs, respectively, are shown as examples. Numberings on (B, B', C, C', C* and C*') indicate GSCs. (C*) Another optical section of (C) for indicating the GSCs residing deeper location. (B', C', and C*') Magenta channels of (B, C, and C*). Scale bar in (B) is 10 μm, and applicable for (C and C*). (D-F) Bar graphs overlaid with beeswarm plots represent numbers (No.) of GSCs per germarium. Genotypes (D), *Gal4* lines driving *UAS-Ptp10D.IR*[GD115] (lane 1–5 in E) and *UAS-sas.IR* (lane 6–10 in E), and temperature-shift conditions in indicated flies (F) are shown. *Dcr-2* was used to enhance RNAi efficiency. Data values indicated by 18°C in (F) are from germaria at 2 days after eclosion without mating

(corresponding to 1 day after eclosion on 29˚C) and all others in (D-F) are from germaria at 1 day after eclosion without mating. P-values (black) (***P < 0.0001, *P<0.01, 0.01 < "actual P-value" < 0.05) for Wilcoxon rank sum test are shown. P-value (red) for Kruskal-Wallis rank sum test to "lanes 1–4" (red line in D) are shown. P-values (red) (***P < 0.0001) for Mann-Whitney U test with Bonferroni correction (post-hoc test) are shown.

## Results

### Loss of *Ptp10D* causes excessive GSCs in the female germarium

Polarity-deficient mutant cells emerged in *Drosophila* epithelium are eliminated by apoptosis when surrounded by wild-type cells, a phenomenon called tumor-suppressive cell competition [18,22,23]. We have previously shown that tumor-suppressive cell competition is driven by the apical transmembrane protein Sas and its receptor Ptp10D, a type III receptor-type protein tyrosine phosphatase (RPTP), which inhibits Egfr signaling [20]. However, the physiological role of the cell competition machinery in normal development is still elusive. While *sas* mutant flies are larval lethal [20,24], an amorphic allele of *Ptp10D*, *Ptp10D¹*, is homozygous viable and fertile without showing any obvious abnormalities in external morphology [25,26]. This led us to investigate the detailed phenotype of *Ptp10D¹* mutant flies to dissect the physiological role of the cell competition machinery. We extensively analyzed epidermis, eyes, appendages, bristles, digestive system, and the reproductive system of *Ptp10D¹* mutant flies and found that the female germaria (Fig 1A) of *Ptp10D¹* flies possess significantly increased number of GSCs that are marked by the anti-phosphorylated Mad (p-Mad) and anti-Hts (which labels a unique intracellular structure called spectrosome in GSCs) [6,27] (Fig 1B, 1C and 1C* and S1 and S2 Movies). Whereas wild-type germarium constantly contains two or three GSCs (Fig 1D) [6], *Ptp10D¹* flies possessed a two-fold number of GSCs (Fig 1D). The trans-heterozygote of *Ptp10D¹* and *Df(1)Exel6242*, a chromosomal deficiency that deletes the *Ptp10D* gene locus, or *Ptp10D^EP1172*, a hypomorphic allele of *Ptp10D* [26], also exhibited a similar excessive GSCs phenotype (Fig 1D), indicating that *Ptp10D* is responsible for the excessive GSCs phenotype in *Ptp10D¹* flies. This increase in the number of GSCs in *Ptp10D¹* flies was significantly suppressed by deleting one copy of the *Egfr* gene (Fig 1D), which is consistent with the previous data that Ptp10D drives trachea formation and cell competition by negatively regulating Egfr signaling [20,28]. Notably, loss of another *Drosophila* type III RPTP *Protein tyrosine phosphatase 4E* (*Ptp4E*), which genetically interacts with *Ptp10D* in axon guidance and tracheal formation [21,28], did not cause the excessive GSCs phenotype (Fig 1D), similarly to the previous data that Ptp10D but not Ptp4E drives cell competition [20].

### Ptp10D and Sas in gonadal apical cells determine the number of GSCs

We next examined which cells are responsible for excessive GSCs in *Ptp10D* mutant flies. Using the Gal4/UAS system, we performed a cell type-specific knockdown of *Ptp10D* or its ligand *sas*. Unexpectedly, knockdown of *Ptp10D* or *sas* in germ cells using the *nos.NGT-Gal4* (S1A and S1E Fig) did not cause excessive GSCs (Fig 1E). Instead, knockdown of *Ptp10D* or *sas* using a somatic cell-specific *Gal4*-driver *C587-Gal4* [29,30] (S1B and S1F Fig) caused significantly increased GSCs (Fig 1E). These data suggest that Ptp10D and Sas act in somatic cells, but not germ cells, in the regulation of GSC number in the germarium.

To clarify the responsible somatic cells that determine the number of GSCs, we further knocked down *Ptp10D* or *sas* using the *bab1-Gal4*, which drives gene expression at the larval stage in apical cells, terminal filament cells, and swarm cells, as well as at the adult stage in terminal filament cells and cap cells [31,32] (S1C and S1G Fig) or the *tj-Gal4*, which acts at the larval stage in terminal filament cells, interstitial cells, and swarm cells, as well as at the adult stage in escort cells [32] (S1D and S1H Fig). As a result, we found that knockdown of *Ptp10D* or *sas* using the *bab1-Gal4* but not *tj-Gal4* caused excessive GSCs (Fig 1E). This phenotype

caused by *Ptp10D* knockdown was further confirmed by using two independent *UAS-RNAi* lines against *Ptp10D*, *GD115* (Vienna Drosophila RNAi center) (Fig 1E) and *TRiP.HMS01917* (Drosophila RNAi Screening Center) (S2A–S2C Fig). These data indicate that apical cells at the larval stage or terminal filament cells and cap cells at the adult stage could be the responsible cells. Thus, we further sought to determine the developmental stage at which *Ptp10D* or *sas* acts in the regulation of GSC number by performing a stage-specific knockdown using the TARGET (temporal and regional gene expression targeting) system, a temperature-dependent induction of the *Gal4-UAS* system using the temperature-sensitive Gal4 repressor Gal80 (Gal80$^{ts}$) [33]. Larvae bearing the transgenes *bab1-Gal4*, *UAS-Ptp10D.IR*, and *tub-Gal80$^{ts}$* were raised at permissive temperature (18˚C, RNAi-OFF) and then sifted to restrictive temperature (29˚C, RNAi-ON) at specific stages (one day before wandering stage (1d BW), wandering stage, and one day after puparium formation (1d APF)) (Fig 1F). A temperature shift before the puparium formation caused excessive GSCs (1d BW, P < 0.0001 and wandering, P < 0.01; Fig 1F), while a temperature shift after that did not affect the number of GSCs (1d APF; Fig 1E), suggesting that the pupal/larval transition (wandering L3) stage is the critical period for Ptp10D to control the number of GSCs. Although knockdown of *sas* using the same TARGET system significantly reduced the eclosion rate, escapers showed a similar excessive GSCs phenotype that was observed by *Ptp10D* knockdown (Fig 1F). Taken together, these data indicate that gonadal apical cells at the wandering L3 stage (Fig 1A, center) are the responsible cells that determine the number of GSCs via Sas-Ptp10D signaling.

## Ptp10D and Sas in gonadal apical cells control the shape of the GSC niche

We next investigated the mechanism by which loss of *Ptp10D* or *sas* in gonadal apical cells causes excessive GSCs. Intriguingly, we found that *Ptp10D$^{1}$* germarium show abnormal shape in the cluster of cap cells (Fig 2A–2C, arrowheads), which act as niche cells for GSCs [5,7]. In wild-type germarium, the cap cell-cluster shapes a dish-like structure that is composed of Engrailed (En)-positive cells with F-actin accumulation asymmetrically spreading at the tip of the ovarian sheath (Fig 2A and 2G). This structure allows only two or three GSCs to be able to physically interact with the niche [5]. Notably, the shape of the cap cell-cluster in *Ptp10D$^{1}$* was not dish-like but aggregated or protruded into the ovarian space (Fig 2B, 2C and 2G). A cross-sectional confocal analysis revealed that this abnormal niche structure allows more GSCs to physically interact with the cap cell-cluster (S3 Fig), suggesting that the increased GSCs in *Ptp10D$^{1}$* germarium is due to an increased capacity of the niche to harbor GSCs. We further examined whether morphological changes of cap cells occurred in *Ptp10D$^{1}$*. However, we did not found alterations in differentiation (S6A–S6D Fig), cell adhesion (S4A and S4B Fig), and actin-cytoskeleton (S4C and S4D Fig). We also found that *Ptp10D$^{1}$* flies did not show defects in the structure of muscle sheath (S4E and S4F Fig), the primordial cells of which are gonadal apical cells (Fig 1A) [9]. We only found that the number of cap cells was slightly (approximately 1 cell) increased in *Ptp10D$^{1}$* (S4G Fig). However, it is likely that one-cell increase in the number of cap cells does not have significant impact on expanding the surface of cap cell-cluster. Stage-specific knockdown of *Ptp10D* (Figs 2E, 2G, and S2D–S2F) or *sas* (Fig 2F and 2G) in gonadal apical cells using *bab1-Gal4* also caused abnormal shape of the cap cell-cluster. Consistent with these data, Ptp10D and Sas were broadly expressed in gonadal apical cells as visualized by anti-Ptp10D antibody and GFP-tagged Sas (Sas::SGFP) [34] (Fig 2H, 2H', and 2H"). We also confirmed that *Ptp10D$^{1}$* abolished Ptp10D protein expression in gonads and that *bab1*-Gal4-mediated knockdown of *Ptp10D* or *sas* reduces Ptp10D protein expression or Sas:: SGFP in apical cells, respectively (S5 Fig). These results suggest that Ptp10D and Sas in gonadal apical cells control the shape of the niche for GSCs during germarium development.

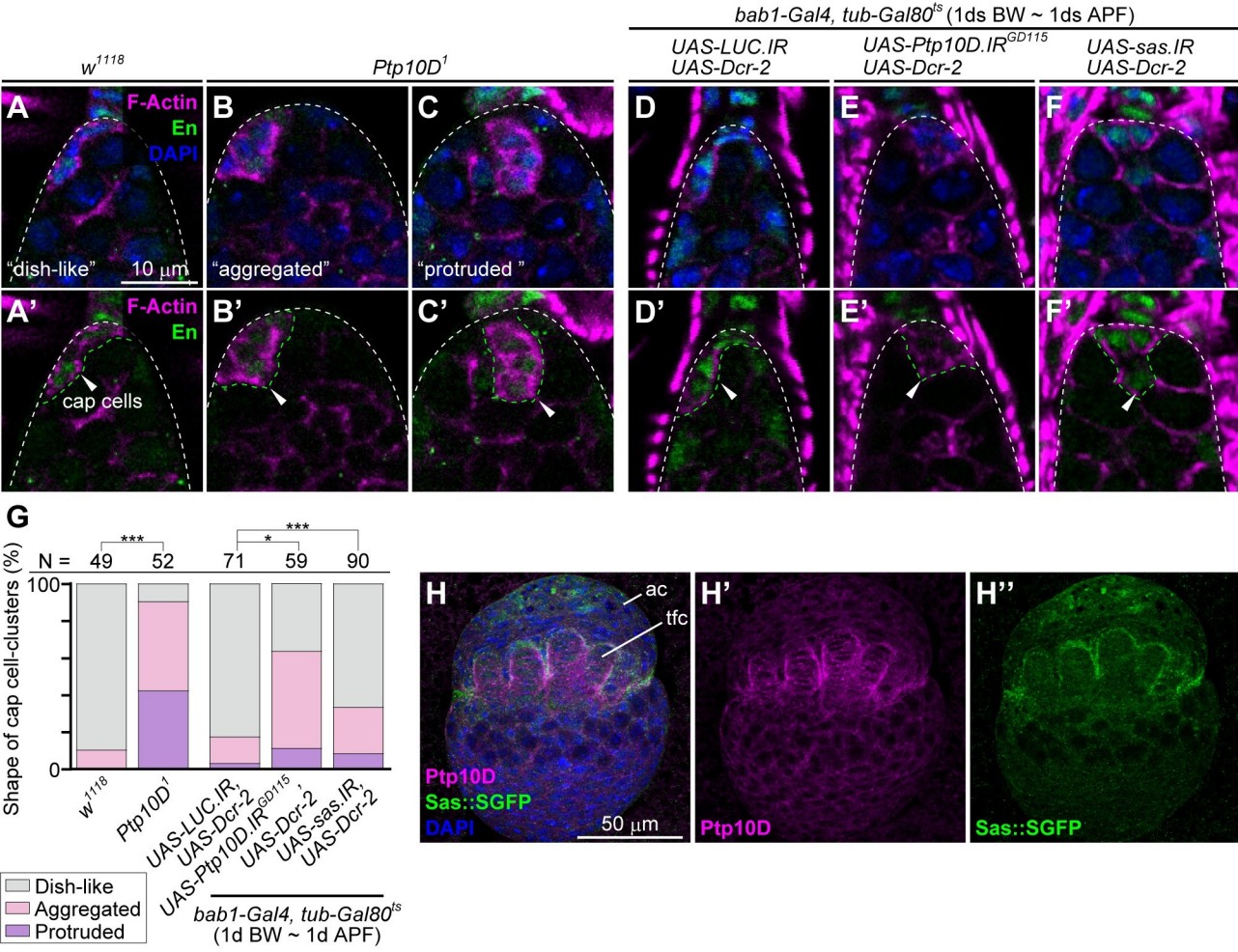

**Fig 2. Loss of *Ptp10D* or *sas* in gonadal apical cells causes abnormal shaping of cap cell-cluster.** (A-G) Proximal regions of female germaria without mating labeled with Phalloidin (magenta), anti-En antibody (green), and DAPI (blue). Proximal is to the top. (A-C) Germaria at 1 day after eclosion are shown. *w¹¹¹⁸* homozygote with "dish-like" cap cell-cluster (A), and *Ptp10D¹* homozygotes with "aggregated" (moderately abnormal) (B) and "protruded" (severely abnormal) (C) ones, respectively are shown as examples. (D-F) Germaria of indicated genotypes 2 days after eclosion that are treated with the temperature-sift (29°C, 1d BW ~ 1d APF) are shown. Germaria with the abnormal cap cell-clusters are shown as examples. *LUC.IR* was used as a control. *Dcr-2* was used to enhance RNAi efficiency. (A'-F') Magenta/green channels of (A-F). White dashed lines in (A-F) and green dashed line with arrowheads in (A'-F') indicate outlines of germaria and cap cell-clusters, respectively. (G) 100%-stacked bar graph represents % of germaria with normal (gray), aggregated (pink), and protruded (purple) cap cell-clusters in indicated genotypes. Numbers (N) of samples observed and P-value (***P < 0.0001, *P<0.01) for Fisher's exact test are shown. (H) Female gonad of late wandering larva which contains two copy of Sas::SGFP transgene was labeled with anti-Ptp10D antibody (magenta), SGFP fluorescence/anti-GFP antibody (green) and DAPI (blue). The cell layer located at a opposite side of fat body adherent surface are shown. (H' and H") Magenta and green channels of (H), respectively. ac: apical cells, tfc: terminal filament cells. Scale bar in (H) is 50 μm. Ptp10D is expressed ubiquitously and exhibits strong expression at the boundary between apical cells and terminal filament cells. Sas::SGFP shows higher expression in apical cells, especially in the most-distal region, compared to other cell types at basal half of gonad.

## Ptp10D controls the shape of GSC niche via Egfr signaling

Given that the excessive GSCs phenotype in *Ptp10D¹* was suppressed by the reduction of *Egfr*, the abnormal shape of the cap cell-cluster in *Ptp10D¹* could be caused by elevated Egfr signaling in gonadal apical cells. Indeed, *Ptp10D¹* exhibited elevated Egfr signaling in gonadal apical cells, as visualized by anti-phospho-ERK (pERK) staining (Fig 3A–3C), and stage-specific knockdown of *Egfr* in gonadal apical cells by *bab1-Gal4* significantly suppressed the abnormal shape of the cap cell-cluster in *Ptp10D¹* (Fig 3D). Therefore, we examined the activity of Rolled/ERK, a MAP kinase downstream of Egfr, using the *tub*-miniCic::mCherry (Fig 3E and 3E'), a highly-

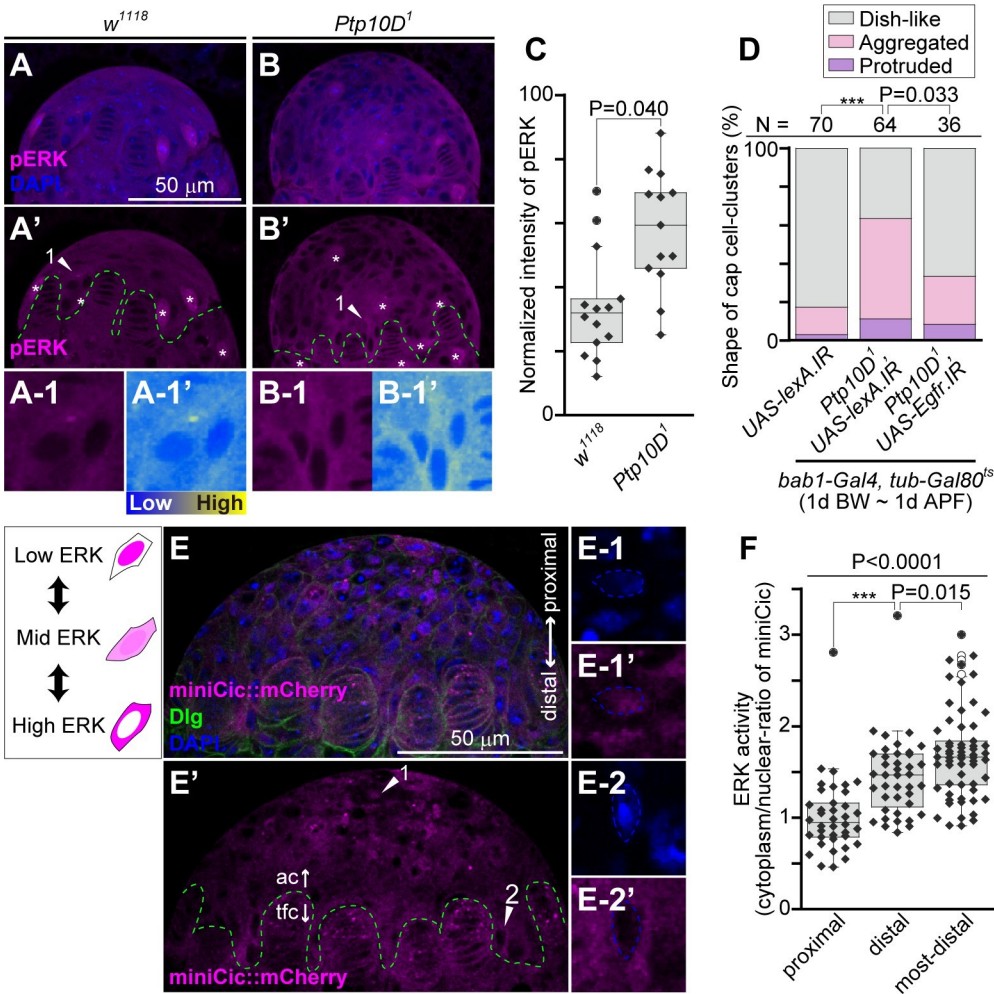

**Fig 3. Loss of *Ptp10D* causes abnormal shape of cap cell-cluster via elevated Egfr signaling.** (A and B) Female gonads of late wandering larvae in *w^1118^* and *Ptp10D^1^* homozygotes were labelled with anti-pERK antibody (magenta) and DAPI (blue). (A' and B') Magenta channels of (A and B). Asterisks in (A' and B') indicate non-specific staining of anti-pERK in mitotic cells. Green dashed lines in (A' and B') indicate the boundaries between apical cells and terminal filament cells. Apical cells indicated by arrowheads 1 in (A' and B) are shown in (A-1 and B-1). Heatmap images of (A-1 and B-1) are shown in (A-1' and B-1'). (C) Boxplot overlaid with beeswarm plot represents the normalized intensity of pERK staining. Data values from *w^1118^* and *Ptp10D^1^* are plotted. P-value for Mann-Whitney U test are shown. (D) 100%-stacked bar graph represents % of germaria with normal (gray), aggregated (pink), and protruded (purple) cap cell-clusters in indicated genotypes. *lexA.IR* was used as a control. Numbers (N) of samples observed and P-values (***P < 0.0001, 0.01 < "actual P-value" < 0.05) for the pairwise comparison using Fisher's exact test with Bonferroni correction are shown. P-values for the pairwise comparison to the dish-like/aggregated pair are shown. Other pair (dish-like/protruded and aggregated/protruded) did not showed significant differences (P < 0.05) for this pairwise comparison. (E) Female gonads of late wandering larvae having one copy of *tub-miniCic::mCherry* transgene were labeled with mCherry fluorescence (magenta) and anti-mCherry antibody (magenta), anti-Dlg antibody (green) and DAPI (blue). (E') Magenta channel of (E). Green dashed line in (E') indicates the boundary between apical cells (ac) and terminal filament cells (tfc). A proximal cell and a most-distal cell exhibiting typical ERK activity indicated by arrowheads 1 and 2 in (E') are shown in (E-1 and E-1') and (E-2, E-2'), respectively. Blue channels (E-1 and E-2) and magenta channels (E-2 and E-2') are shown. Blue dashed lines in (E-1' and E-2') indicate outlines of nuclei. (F) Boxplot overlaid with beeswarm plot represents the ratio of mimiCic intensity in cytoplasm to that in nucleus in each apical cell. Data values from apical cells located in proximal region, distal region, and most-distal region are plotted. P-value for Kruskal-Wallis rank sum test to three distributions is shown (line). P-value (***P < 0.0001, 0.01 < "actual P-value" < 0.05) for Mann-Whitney U test with Bonferroni correction (post-hoc test) are shown.

sensitive reporter for the ERK activity [35]. We found that the distal population of apical cells exhibited high ERK activity, while the proximal population exhibited low ERK activity (Fig 3E, 3E', and 3F). Notably, ERK activity of the most-distal apical cells nearby terminal filaments was extremely high, as they lost the nuclear localization of the miniCic::mCherry reporter (Fig 3E-2, 3E-2', and 3F). This pattern of ERK activity implies that the apical cells nearby terminal filaments are involved in the shaping of the cap cell-cluster.

## Blocking apoptosis in apical cells phenocopies loss of *Ptp10D* or *sas* in shaping the cap cell-cluster

The previous data that Sas-Ptp10D signaling promotes apoptosis in tumor-suppressive cell competition [20] suggests that the abnormal shape of the cap cell-cluster in *Ptp10D[1]* germaria is due to suppression of apoptosis. To test this possibility, apoptosis was genetically blocked by overexpression of *RHG.miRNA*, an artificial microRNA that simultaneously suppresses *Drosophila* proapoptotic genes *reaper*, *head involution defective* (*hid*) and *grim* [36], baculovirus *p35*, an inhibitor of effector caspases [37], or *Death-associated inhibitor of apoptosis 1* (*Diap1*), an inhibitor of Caspase [38]. The stage-specific overexpression of *RHG.miRNA* (Fig 4B and 4C), *p35* (Fig 4C), or *Diap1* (Fig 4C) in gonadal apical cells by *bab1-Gal4* caused abnormal shape of the cap cell-cluster (Fig 4A, 4B and 4C) accompanied with the increased GSCs (S6 Fig), which phenocopies loss of *Ptp10D* or *sas* (Fig 2E–2G). These results suggest that apoptosis in gonadal apical cells contributes to proper shaping of the cap cell-cluster.

## *Ptp10D[1]* gonads have reduced apoptosis in the apical cells via Egfr signaling

We thus examined whether gonadal apical cells indeed undergo apoptosis via Ptp10D. During female gonadogenesis, part of apical cells differentiates into terminal filament cells at the mid-third larval stage [9,11] and then terminal filament cells form rows by the wandering-third larval stage (Fig 1A). We found that apoptotic cells stochastically emerge in the proximal population of the apical cells at the late-third larval stage, as visualized by the anti-cleaved caspase (c-Dcp-1) antibody (S7A and S7B Fig). Subsequently, the areas of apoptosis expands to the distal region of the apical cells nearby terminal filament cells at the wandering-third larval stage (S7C and S7D Fig), the time point at which the apical cells move to separate rows of terminal filament cells (Fig 1A) [9,11]. Notably, apoptosis of the apical cells, particularly those close to terminal filament cells, was significantly suppressed in *Ptp10D[1]* gonads (Fig 4D, 4E and 4G). Furthermore, the reduced apoptosis in *Ptp10D[1]* gonads was suppressed by knockdown of *Egfr* in the apical cells using *bab1-Gal4* (Fig 4D–4F and 4G). These data suggest that Ptp10D negatively regulates Egfr to facilitate apoptosis in the gonadal apical cells to properly shape the cap cell-cluster.

We also confirmed that overexpression of *RHG.miRNA* or *Diap1* significantly reduced the number of c-Dcp-1-positive apical cells adjacent to the terminal filament cells, while overexpression of *p35* only showed a tendency to suppress apoptosis (S8 Fig). The moderate suppression of apoptosis by *RHG.miRNA*, *p35*, or *Diap1* was consistent with the moderate germarium phenotypes (Figs 4C and S8D). These results support the notion that suppression of apoptosis in the larval gonad correlates with abnormalities in the adult germarium.

## *Ptp10D[1]* gonads have reduced JNK activity in the apical cells via Egfr signaling

We next sought to identify the signaling pathway that links reduced Egfr signaling to apoptosis in shaping the cap cell-cluster. It has been shown that elimination of polarity-deficient cells by

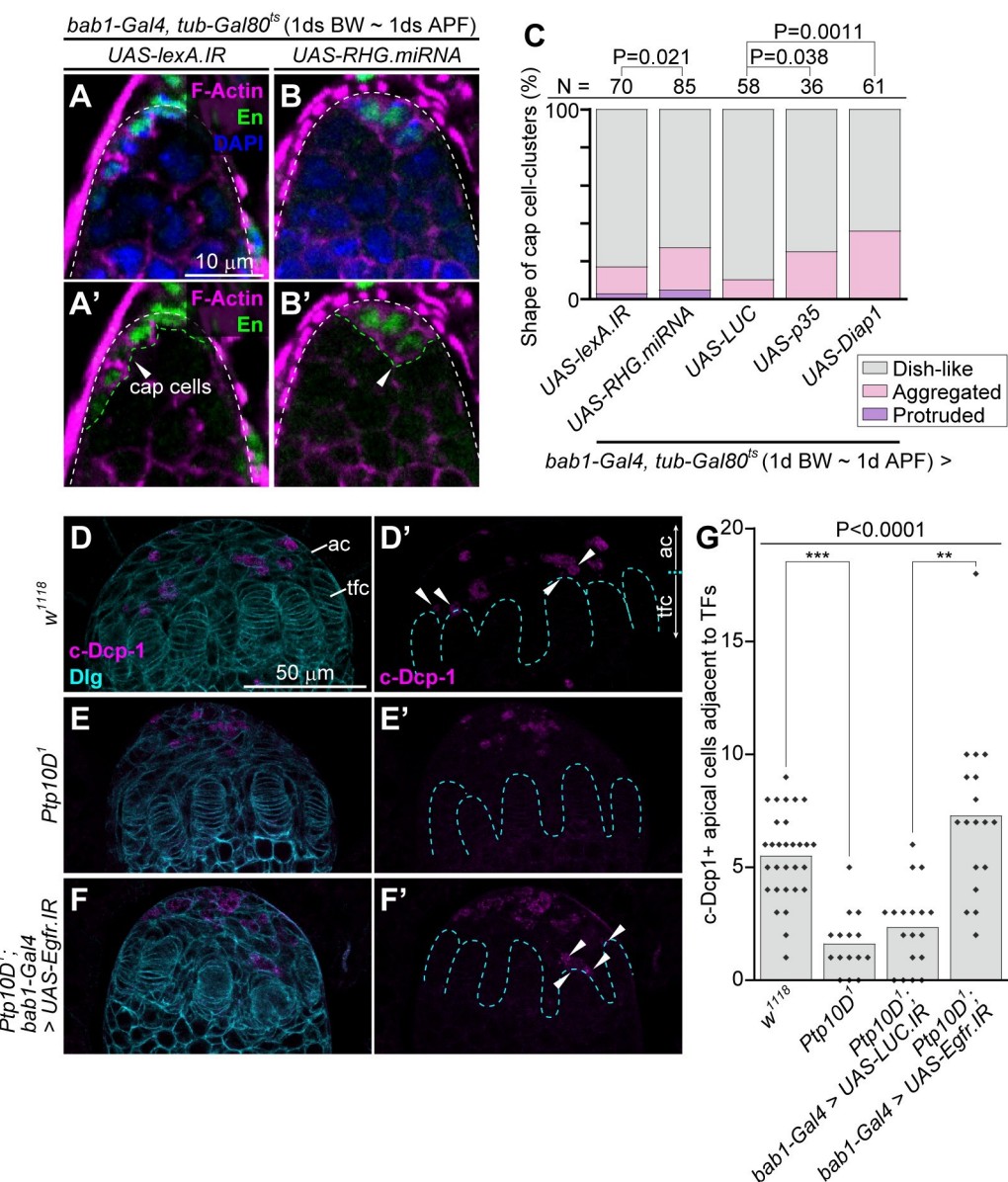

**Fig 4. *Ptp10D[1]* causes the suppression of apoptosis in the apical cells.** (A and B) Germaria of indicated genotypes (upper) 2 days after eclosion that are treated with the temperature-sift (29˚C, 1d BW ~ 1d APF) are shown. Germaria with dish-like (A) and abnormal (B) cap cell-clusters are shown as examples. (A' and B') Magenta/green channels of (A and B). White dashed lines in (A, A', B, and B') and green dashed line with arrowheads in (A' and B') indicate outlines of germaria and cap cell-clusters, respectively. Scale bar in (A) is 10 μm, and applicable for (A', B and B'). (C) 100%-stacked bar graph represents % of germaria with normal (gray), aggregated (pink), and protruded (purple) cap cell-clusters in indicated genotypes. *lexA.IR* (use same data value in Fig 3A, lane 1) and *LUC* are used as controls. Numbers (N) of samples observed and P-value for Fisher's exact test are shown at the upper. (D-F) Female gonads at wandering L3 stage (corresponding to the stage of S7D Fig) are labeled with anti-c-Dcp-1 antibody (magenta) and anti-Dlg antibody (cyan). The cell layers located at a opposite side of fat body-adherent surface are shown. Genotypes of samples are indicated at the left. Images are processed by the Z-stack projection of two sections (corresponding to 3 μm thickness) to visualize the boundary between terminal filament cells. (D'-F') Magenta channels of (D-F). Scale bar in (D) is 50 μm, and applicable for (E, F, and D'-F'). Cyan dashed lines in (D'-F') indicate boundary between apical cells (ac) and terminal filament cells (tfc). Arrowheads in (D' and F') indicate the c-Dcp-1-positive apical cells which adjacent to terminal filaments. (G) Bar graph overlaid with beeswarm plots represent numbers (No.) of c-Dcp-1-positive apical cells adjacent to terminal filaments (TFs) per larval gonads in indicated genotypes. *LUC.IR* was used as control. P-value for Kruskal-Wallis rank sum test are shown. P-values (***P < 0.0001, **P < 0.001) for Mann-Whitney U test with Bonferroni correction (post-hoc test) are shown.

cell competition via Sas-Ptp10D signaling is driven by JNK-dependent apoptosis [20]. Strikingly, inhibition of JNK signaling by overexpression of a JNK phosphatase *puckered* (*puc*) resulted in abnormal shape of the cap cell-cluster (Fig 5B and 5C) and reduced apoptosis in gonadal apical cells (Fig 5D and 5E). We thus analyzed JNK activity in gonads using a newly generated *puc*-Stinger reporter, a *puc* enhancer reporter that visualizes JNK activity with high sensitivity (S9 Fig). In wild-type gonads, the *puc*-Stinger signal was detectable in the apical cells at the mid-third larval stage (S10A and S10A" Fig) and was enhanced at the late-third larval stage in the proximal population of the apical cells S10B, S10B", S10C and S10C" Fig), where apoptosis begins to emerge (S10C' Fig), followed by a gradual reduction in the signal during the wandering-third larval stage (S10D, S10D", S10E and S10E" Fig). Significantly, *Ptp10D*$^1$ gonads showed reduced *puc*-Stinger signal in the apical cells at the late-third and wandering-third larval stages (Fig 5G, compare to Fig 5F, quantified in Fig 5I). Furthermore, the reduction in the *puc*-Stinger signal in *Ptp10D*$^1$ gonads was cancelled by knockdown of *Egfr* (Fig 5H and 5I). Taken together, these data indicate that Ptp10D negatively regulates Egfr to facilitate JNK-mediated apoptosis in the apical cells.

## Loss of *Ptp10D* in the gonadal apical cells diminishes egg production

Our data presented so far show that loss of Ptp10D reduces JNK-mediated apoptosis in the gonadal apical cells, which results in abnormal shaping of the GSC-niche and thus increased number of GSCs. We finally sought to address the phenotypic outcome of the excessive GSCs. To address this, we measured the number of egg laying (S11 Fig) and ovarioles per female and calculated the number of eggs per ovariole as a reproductive capacity (Fig 6). We found that, while heterozygous females for *Ptp10D*$^1$ produced 6–7 eggs per ovarioles in a week, homozygous females for *Ptp10D*$^1$ produced only 2–3 eggs per ovarioles (Fig 6A and 6A'). We also found that trans-heterozygous females for *Ptp10D*$^1$ and *Ptp10D*$^{EP1172}$, which moderately increased the number of GSCs (Fig 1D), exhibited intermediate phenotype in egg production (Fig 6A), suggesting that increased GSCs number correlates with decreased egg laying. Furthermore, stage-specific knockdown of *Ptp10D* in gonadal apical cells by *bab1-Gal4* also resulted in diminished egg production per ovariole (Fig 6B and 6B'). These data suggest that Ptp10D in the apical cells contributes to the production of larger number of eggs by properly shaping the cap cell-cluster that keeps 2–3 GSCs per germarium. *Ptp10D*$^1$ females did not show increased apoptosis in germ cells during oogenesis in germarium or oocytes (S12 Fig).

## Discussion

In this study, we have shown that the Sas-Ptp10D machinery facilitates JNK-mediated apoptosis of gonadal apical cells to properly shape the niche structure of germ-line stem cells (Fig 7). In cell competition between polarity-deficient clones and wild-type clones, Sas-Ptp10D facilitates JNK-mediated apoptosis of polarity-deficient cells at the clone boundary [20]. Similar to cell competition, our present data show that Sas-Ptp10D facilitates apoptosis of the apical cells at the boundary between apical cells and terminal filament cells (Figs 4D–4G and S7D). The mechanism by which Sas-Ptp10D regulates apoptosis only in the apical cells nearby terminal filament cells is currently unknown. Notably, Egfr-ERK signaling is significantly elevated in living apical cells nearby terminal filament cells during normal development (Fig 3E and 3F), which suggests that apical cells in this region survive in an Egfr-ERK-dependent manner and that *Ptp10D* deficiency increases Egfr-ERK signaling and thus promotes their survival. On the other hand, the mechanism by which apoptosis of apical cells ensures the proper shaping of cap cell-cluster is still unclear. Notably, we found that suppression of apoptosis and misshaping of cap cell-cluster are correlated (Figs 4C–4G, 5D, 5E, and S8). It may be possible that

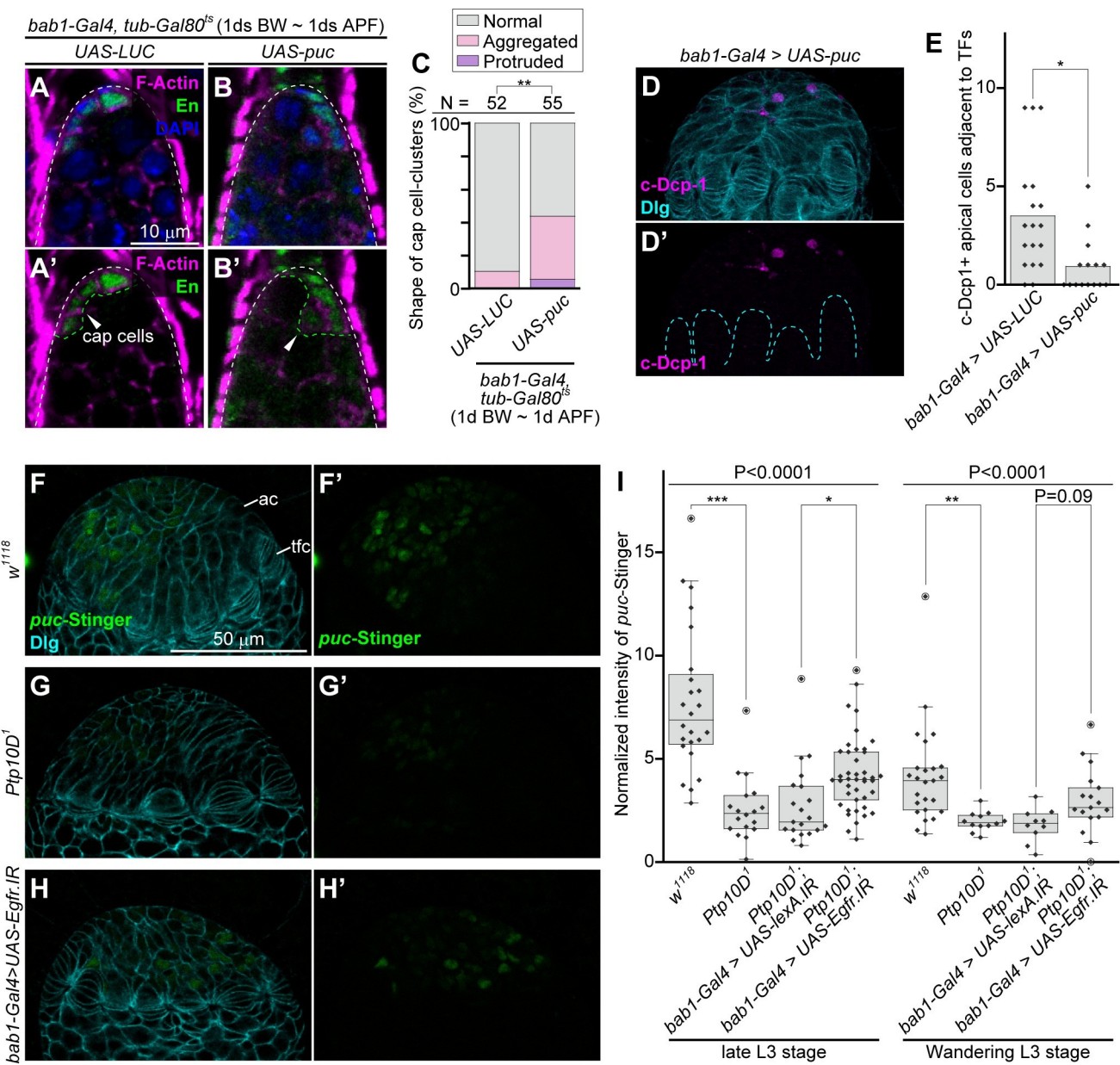

**Fig 5. *Ptp10D¹* causes the reduction of JNK signaling activity in the apical cells.** (A and B) Germaria of indicated genotypes (upper) 2 days after eclosion that are treated with the temperature-sift (29°C, 1d BW ~ 1d APF) are shown. Germaria with dish-like (A) and abnormal (B) cap cell-clusters are shown as examples. (A' and B') Magenta/green channels of (A and B). White dashed lines in (A, A', B, and B') and green dashed line with arrowheads in (A' and B') indicate outlines of germaria and cap cell-clusters, respectively. Scale bar in (A) is 10 μm, and applicable for (A', B and B'). (C) 100%-stacked bar graph represents % of germaria with normal (gray), aggregated (pink), and protruded (purple) cap cell-clusters in indicated genotypes. *LUC* (use same data value in Fig 4C, lane 3) is used as a control. Numbers (N) of samples observed and P-value for Fisher's exact test are shown at the upper. (D) Female gonad at wandering L3 stage (corresponding to the stage of S7D Fig) are labeled with anti-c-Dcp-1 antibody (magenta) and anti-Dlg antibody (cyan). The cell layer located at a opposite side of fat body-adherent surface is shown. Genotypes of samples are indicated at the upper. Images are processed by the Z-stack projection of two sections (corresponding to 3 μm thickness) to visualize the boundary between terminal filament cells. (D) Magenta channels of (D). Scale bar in (D) is 50 μm, and applicable for (D'). Cyan dashed lines in (D') indicate boundary between apical cells (ac) and terminal filament cells (tfc). (E) Bar graph overlaid with beeswarm plots represent numbers (No.) of c-Dcp-1-positive apical cells adjacent to terminal filaments (TFs) per larval gonads in indicated genotypes. *LUC* was used as control. P-value (*P < 0.01) for Mann-Whitney U test is shown at the upper. (F-H) Female gonads at late L3 stage (corresponding to the stage of S10C Fig) are labeled with *puc*-Stinger fluorescence (green) and anti-Dlg antibody (cyan). Genotypes of samples are indicated at the left. The cell layers located at a opposite side of fat body-adherent surface are shown. (F'-H') Magenta channels of (F-H). Scale bar in (F) is 50 μm, and applicable for (G, H, and F'-H'). (I) Boxplot overlaid with beeswarm plot represents normalized intensity of *puc*-Stinger in indicated genotypes. *lexA.IR* was used as a control. Data value from late L3 stage (corresponding to the stage of S6C Fig) and wandering L3 stage (corresponding to the stage of S6E Fig) are shown. P-values for Kruskal-Wallis rank sum test to four distributions (lane 1–4 and lane 5–8) are shown at the upper (line). P-values (***P < 0.0001, **P < 0.001, *P < 0.01, 0.01 < "actual P-value" < 0.05) for Mann-Whitney U test with Bonferroni correction (post-hoc test) are shown.

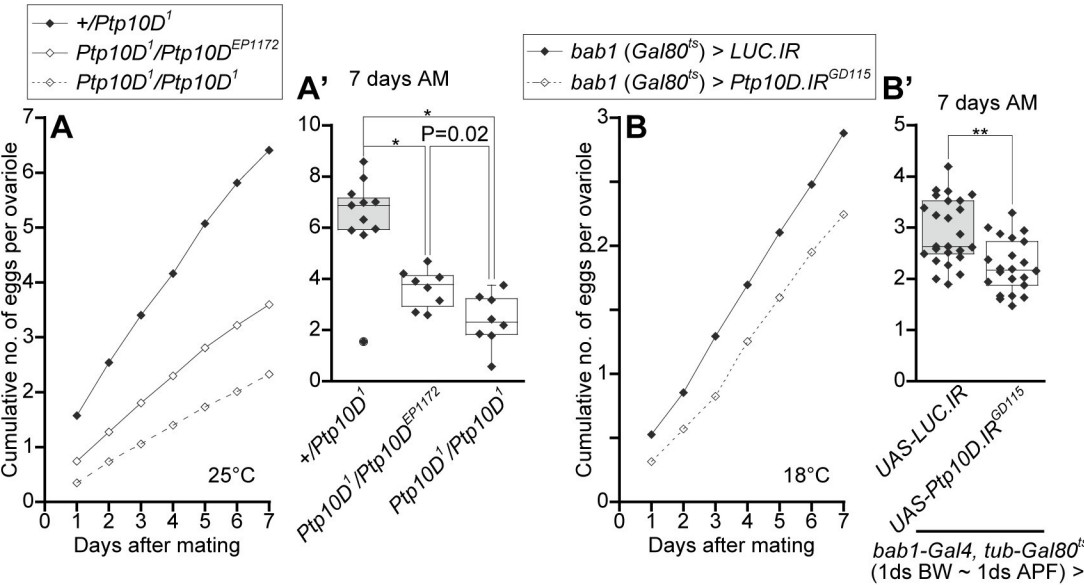

**Fig 6. Loss of *Ptp10D* diminishes the efficient egg production.** (A and B) Line graphs represent means of cumulative numbers of eggs per ovarioles laid by single females after mating with single wild-type *Canton-S* males. (A) Data values from control *+/Ptp10D¹* (black line and closed dots in A), *Ptp10D¹/Ptp10D^EP1172* (black line and open dots in A), and *Ptp10D¹/Ptp10D¹* (dashed line and open dots in A) at 25 degrees. (B) Data values from control *UAS-LUC.IR* (black line and closed dots in B) and *UAS-Ptp10D.IR* (dashed line and open dots in B) by *bab1-Gal4*/TARGET at 18 degrees. (A' and B') Boxplots overlaid with beeswarm plots representing row data value of 7 days after mating (AM) in (A and B). P-values (**P < 0.001, *P<0.01) for Mann-Whitney U test are shown.

apoptosis of apical cells contributes to appropriate migration of apical cells. Given that the migration of apical cells along the proximo-distal axis individualizes the cluster of terminal filament cells and forms the outline of each germalium unit (Fig 1A) [12], the appropriate migration of cap cells would contribute to proper shaping of cap cell-cluster.

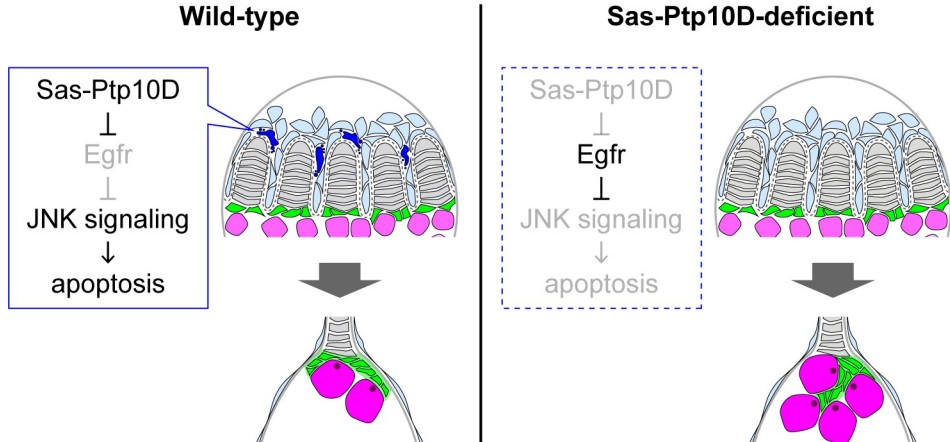

**Fig 7. Hypothesis for the role of Sas-Ptp10D in controlling the GSCs via shaping GSC-niche.** In the gonadogenesis at pre-pupal stage (left), apoptosis is stochastically emerged in apical cells neighboring terminal filament cells. The Sas-Ptp10D-mediated apoptosis of apical cells ensure shaping the dish-like structure of cap cell-cluster which restrict the number of GSCs to 2 or 3. Sas-Ptp10D has roles to facilitate apoptosis by downregulation of the Egfr signaling, which suppressed the activation of the JNK signaling. In Sas-Ptp10D-deficient state (right), hyperactivation of the Egfr signaling suppressed the JNK signaling-inducible apoptosis. Suppression of apoptosis caused disruption of appropriate shaping of cap cell-cluster, thereby the residential space (niche) of GSCs is expanded to increase the number of GSCs.

Our data show that downregulation of Egfr activity by Sas-Ptp10D facilitates apical cell elimination via JNK-mediated apoptosis. Similar life-or-death decisions by the ERK-JNK balance have been shown in mammalian systems [39,40]. A possible downstream molecule by which the ERK-JNK balance regulates apical cell death could be a *Drosophila* initiator caspase Dronc [38,41]. It has been shown that Dronc activates JNK signaling during tissue regeneration [42] and that JNK positively regulates caspase cascade [43,44], forming an amplification loop between Dronc and JNK [45]. Given that Egfr-ERK signaling negatively regulates the activity of Hid [46,47], which activates Dronc, Egfr-ERK may antagonize JNK-mediated apoptosis via inhibition of the Hid-Dronc-JNK amplification loop. Thus, a negative regulator of Egfr such as Sas-Ptp10D could act as a key regulator that determines life-or-death in the gonadal apical cells (Fig 7).

Our data show that *Ptp10D* homozygous mutant flies possess excessive GSCs in female germaria (Fig 1B–1D). Intriguingly, the increased GSCs did not result in increased reproductive capacity but rather reduced it (Figs 6 and S11). Notably, we did not find any significant reduction in Dpp signaling activity in *Ptp10D* mutant GSCs (Fig 1B and 1C), suggesting that the stemness maintenance in GSCs is not affected. We also did not find any increase in the number of apoptotic cells as a defect in oogenesis in *Ptp10D* mutants (S12 Fig). Instead, the abnormal shape of the cap cell-cluster and increased GSCs in *Ptp10D* mutants may affect the appropriate transition of GSCs from self-renewal to differentiation. It has been reported that GSCs undergo self-renewal-to-differentiation transition by sequentially interacting with cap cells and escort cells in a spatially sophisticated manner [48,49]. The abnormal shaping of the niche and resulting increased GSCs may disrupt the sequential interactions. Our observations propose a concept that the shape of the niche structure optimizes the stem cell system, thereby maximizing the reproductive capacity.

## Materials and methods

### *Drosophila* strains

$w^{1118}$ iso2;3 (Bloomington Drosophila Stock Center (BDSC), #6326), $w^*$ (unspecified *w* allele), and *Canton-S* SNPiso3 (BDSC, #6366) were used as wild-types. *Ptp10D$^1$* (BDSC #5810), *Ptp4E$^1$* (BDSC #42019), and *Egfr$^{f2}$* (BDSC, #2768) were used amorphic alleles for *Ptp10D*, *Ptp4E*, and *Egfr*, respectively. *P{A92}puc$^{E69}$* (*puc$^{E69}$-lacZ*) was used as an enhancer trap line for *puc*. *UAS-mCherry* (BDSC, #52268), *UAS-Dcr-2* on second (BDSC, #24650) and third chromosomes (BDSC, #24651), *UAS-LUC.VALIUM1* (*UAS-LUC*) (BDSC, #35789), *UAS-lacZ.Exel* (*UAS-lacZ*) (BDSC, #8529), *UAS-p35* [50], *UAS-Diap1*, *UAS-puc* [51], and *UAS-eiger$^W$* (*UAS-eiger*) [52] were used for Gal4-driven induction of CDS of *monomeric Cherry fluorescent protein* (*mCherry*), *Dicer-2* (*Dcr-2*), *Photinus pyralis Luciferase* (*LUC*), *Escherichia coli β-galactosidase*, *Autographa californica nucleopolyhedrovirus p35* (*p35*), and *puc*. *UAS-LUC* and *UAS-lacZ* that induces non-*Drosophila* gene was used as a control. *UAS-LUC.IR$^{TRiP.JF01355}$* (*UAS-LUC.IR*) (BDSC, #31603), *UAS-lexA.IR$^{TRiP.HMS05772}$* (*UAS-lexA.IR*) (BDSC, #67947), *UAS-Ptp10D.IR$^{GD115}$* (*UAS-Ptp10D.IR*) (Vienna Drosophila Stock Center (VDRC), #1102), *UAS-Ptp10D.IR$^{TRiP.HMS01917}$* (BDSC, #39001), *UAS-sas.IR$^{GD14869}$* (*UAS-sas.IR*) (VDRC, #39086/discarded stock), and *UAS-Egfr.IR$^{10079R-1}$* (*UAS-Egfr.IR*) (Fly Stocks of National Institute of Genetics, #10079R-1) were used for Gal4-driven induction of inverted repeat (IR) sequences targeting *Photinus pyralis LUC* (*TRiP.JF01355*), *Escherichia coli lexA* (*TRiP.HMS05772*), *Ptp10D* (*GD115*), *Ptp10D* (*TFiP.HMS01917*), *sas* (*GD14869*), and *Egfr* (*10079R-1*). *UAS-LUC.IR* and *UAS-lexA.IR* that targeted non-*Drosophila* genes were used as controls. *GAL4-nos.NGT* (*nos.NGT-Gal4*) (BDSC, #32563), *P{GawB}C587* (*C587-Gal4*) (BDSC, #67747), *P{GawB}bab1$^{Agal4-5}$* (*bab1-Gal4*) (BDSC, #6802), and *P{GawB}NP1624* (*tj-Gal4*) (KYOTO Stock Center, #104055)

were used as cell-type specific *Gal4*-drivers in the female gonad (see S1 Fig for details), and *GMR-Gal4* [53] (a gift from T. Xu) was used as photoreceptor cell-specific *Gal4* driver. *tubP-GAL80^{ts}* (*tub-Gal80^{ts}*) on second (BDSC, #7108) and third chromosomes (BDSC, #7018) were used for TARGET system. *Sas^{15060}::2XTY1-SGFP-V5-preTEV-BLRP-3XFLAG* (*sas::SGFP*) (VDRC, #318129) was used for visualizing the endogenous expression pattern for Sas protein. *Tub-miniCic::mCherry* [35] (a gift from R. Levayer) was used for visualizing the level of ERK activity. *puc-Stinger* was constructed in this report (see S9 Fig). Genotypes of flies used in the experiments are described in S1 File.

## Tissue staining and immunohistochemistry

Ovaries of adult fly were dissected without dissociation of their ovarioles and then fixed with standard 100 mM phosphate-buffered 4% formaldehyde (Paraformaldehyde (nacalai tesque, #26123–55), di-Sodium Hydrogenphosphate 12H$_2$O (nacalai tesque, #31723–35), Sodium Dihydrogenphosphate Dihydrate (nacalai tesque, #31718–15)). For the fixation of female gonads of larval fly with 4% paraformaldehyde, larval body walls were folded inside-out to expose gonads that attach with fat bodies. Immunostaining of fixed samples were washed with PBT (1 x PBS (nacalai tesque, #27575–31), 0.5% TritonX-100 (nacalai tesque, #35501–15)), and then soaked with PBT with 5% Normal Donkey Serum (Jackson ImmunoResearch, #107-000-121) for protein blocking. Samples processed with blocking reagent were washed with PBT and then stained with antibodies. The following primary antibodies and secondary antibodies were used with indicated dilution by PBT: Phospho-Smad1/5 (Ser463/465) (41D10) Rabbit mAb (anti-pMad antibody) (1:50, Cell Signaling Technology (CST), #9516), 1B1-concentrate (anti-Hu-li tai shao (Hts) antibody) (1:50, Developmental Studies Hybridoma Bank (DSHB)), 4D9 anti-engrailed/invected-concentrate (anti-En antibody) (1:100, DSHB), 8B22F5-supernatant (anti-Ptp10D antibody) (1:200, DSHB), anti-Sas antibody [24] (1:2000), LC28.26-supernatant (anti-Lamin C antibody) (1:50, DSHB), DCAD2-concentrate (anti-*D*E-Cadherin antibody) (1:50, DSHB), anti-vasa-supernatant (anti-Vasa antibody) (1:10, DSHB), Phospho-p44/42 MAPK (Erk1/2) (Thr202/Tyr204) (D13.14.4E) XP Rabbit mAb (anti-pERK antibody) (1:100, CST, #4370), 4F3 anti-discs large-concentrate (anti-Discs large 1 (Dlg) antibody) (1:50, DSHB), Anti-Green Fluorescent Protein Antibody (anti-GFP antibody) (1:2000, Aves Labs, #GFP-1020), Anti-mCherry antibody (1:500, Abcam, #ab167453), Cleaved Drosophila Dcp-1 (Asp216) Antibody (anti-c-Dcp-1 antibody) (1:100, CST, #9578), Goat anti-Mouse IgG (H+L) Cross-Adsorbed Secondary Antibody Alexa Fluor 488 (1:250, Invitrogen, #A11001), Goat anti-Mouse IgG (H+L) Highly Cross-Adsorbed Secondary Antibody Alexa Fluor Plus 647 (1:250, Invitrogen, #A32728), Goat anti-Rabbit IgG (H+L) Highly Cross-Adsorbed Secondary Antibody Alexa Fluor 546 (1:250, Invitrogen, #A11035), and Goat anti-Rat IgG (H+L) Cross-Adsorbed Secondary Antibody Alexa Fluor 488 (1:250, Invitrogen, #A11006). For anti-Sas antibody and anti-pERK antibody, Can Get Signal immunostain Solution B (TOYOBO, #NKB-601) and Can Get Signal immunostain Solution A (TOYOBO, #NKB-501), respectively, were used. For labeling F-Actin, samples immunostained were reacted with Alexa Fluor 647 Phalloidin (1:30, Invitrogen, #A22287) for 30 minutes at room temperature. After the immunostaining, samples were washed with PBT, and then soaked in PBT with DAPI (1μg/ml, Sigma-Aldrich, #D9542) for counterstaining of DNA. SlowFade Gold Antifade Mountant with DAPI (Invitrogen, #S36939) or anti-fade mounting medium (1 x PBS, 50% Glycerol (nacalai tesque, #17018–25), 0.2% n-Propyl Gallate (nacalai tesque, #29303–92), referred to https://www.jacksonimmuno.com/technical/products/protocols) were used as mounting reagents for the slide mounting of samples.

## Temporal expression of genes with TARGET system

Females for $w^{1118}/w^{1118}$; tub-Gal80$^{ts}$; bab1-Gal4/TM6B, Tb$^1$ were crossed with males for $w^*/Y$; UAS-Dcr-2; UAS-LUC.IR$^{TRiP.JF01355}$ or $w^*/Y$; UAS-Dcr-2; UAS-Ptp10D.IR$^{GD115}$, and their F1 larvae were reared on 18 degrees, a permissive temperature of Gal80$^{ts}$. For the induction of UAS-targeted sequences from "1 day before wandering stage (1 d BW)", "wandering stage (wandering)", and "1 day after puparium formation (1 d APF)" (Fig 1E), we selected F1 larvae as described below. 1 d BW: F1 larvae without staging were moved from 18 degrees to 29 degrees incubator to rear for 1 day, and wandering larvae were selected to transfer new vials. wandering: wandering larvae before spiracle eversion were selected from F1 larvae without staging to transfer new vials. 1 d APF: wandering larvae exhibiting spiracle eversion were selected from F1 larvae from staging to transfer new vials, and vials were moved to 29 degrees incubator 1 day after the time point when half of larvae became white pupae. After transferring vials from 18 degrees to 29 degrees incubator, Larvae/pupae were reared by 1 day after eclosion and then dissected for immunostaining.

For the induction of UAS-targeted sequences from 1 d BW to 1 d APF with bab1-Gal4/TARGET, F1 larvae without staging were moved from 18 degrees to 29 degrees incubator to rear for 1 day, and wandering larvae were selected to transfer new vials. Selected wandering larvae were reared at 29 degrees incubator when about half of larvae became white pupae. After that, larvae/pupae were further reared at 29 degrees incubator for 1 day and then moved to 19 degrees incubator. After transferring vials from 29 degrees to 18 degrees incubator, larvae/pupae were reared by 2 days after eclosion and then dissected for immunostaining.

## Construction of *puc-Stinger* reporter

Part of the third intron of *puc* (S9A Fig) was amplified with PCR using PrimeSTAR GXL DNA polymerase (Takara Bio, #R050Q) and corresponding primers with 15-bp homologous sequences (small character) to cloning vector (forward primer: atgctgcagcagatcTGATAAGG GTTGTGTCGTCG, reverse primer: gcgcgccggcgaattCCTCTTAGTTAATGACCCCG). The PCR amplicon was subcloned into *pH-Stinger* plasmid [54] digested with *Bgl* II (Takara bio, #1021A) and *EcoR* I (Takara bio, #1040A) using In-Fusion HD Cloning Kit (Takara Bio. #639648) to construct *puc*-Stinger (S9A Fig and S2 File). *puc*-Stinger was injected into *yw* lines by BestGene Inc. and and balanced with *CyO* balancer. The response of *puc-Stinge*r was validated by reference to the expression patters of *puc*$^{E69}$, a *lacZ* enhancer trap of *puc* [55] (S9B and S9C Fig).

## Measurement of apoptotic cells in gonadal apical cells

Images of female gonads stained with anti-c-Dcp-1 antibody, anti-Dlg antibody, and DAPI were acquired using Laser Scanning Confocal Microscope TCS SP8 on DMi8 inverted microscope (Leica Microsystems) controlled with Leica Application Suite X version 2.0.1.14392 (Leica Microsystems). For analysis of apoptosis in apical cells, only female gonads where rows of terminal filament cells were separated (S4D Fig) were selected based on the shape of terminal filaments the outlines of which were visualized with anti-Dlg antibody. Cells labeled with anti-c-Dcp-1 antibody were regarded as apoptotic cells in this experiment for convenience. Apoptotic apical cells, parts of which were contacted to terminal filaments, were regarded as "apoptotic cells adjacent to terminal filaments" and numbers of applicable cells in gonads were manually counted in series of optical-sectional images including whole of apical cell region.

## Measurement of intensity of anti-pERK antibody staining in gonadal apical cells

Images of female gonads stained with anti-Sas antibody and DAPI were acquired using Laser Scanning Confocal Microscope TCS SP8 on DMi8 inverted microscope controlled with Leica Application Suite X. Acquired images were processed with FIJI ImageJ version 1.53f51 [56] for the measurement of anti-pERK antibody staining intensity. The single sections that exhibited strongest intensity were manually selected from series of optical-sectional images capturing apical cell layer located at a opposite side of fat body adherent surface. Mean gray values of apical cell regions which were manually drawn in the selected green channels were measured as "pERK intensity". Mean gray values of cytoplasmic regions of fat body cell which were arbitrary ($> 500$ μm$^2$) drawn in the same section were also measured as "background intensity". Differential values between "pERK intensity" and "background intensity" were calculated as "normalized intensity of pERK".

## Measurement of signal intensity of *puc*-Stinger in gonadal apical cells

Images of female gonads stained with anti-Dlg antibody and DAPI were acquired using Laser Scanning Confocal Microscope TCS SP8 on DMi8 inverted microscope controlled with Leica Application Suite X. Acquired images were processed with FIJI ImageJ version 1.53f51 [56] for the measurement of GFP intensity (one copy of *puc*-Stinger). The single sections that exhibited strongest GFP intensity were manually selected from series of optical-sectional images capturing apical cell layer located at a opposite side of fat body adherent surface. Mean gray values of apical cell regions which were manually drawn in the selected green channels were measured as "GFP intensity". Mean gray values of cytoplasmic regions of fat body cell which were arbitrary ($> 500$ μm$^2$) drawn in the same section were also measured as "background intensity". Differential values between "GFP intensity" and "background intensity" were calculated as "normalized intensity of *puc*-Stinger".

## Analysis of egg production per ovarioles

For the analysis of (Figs 6A, 6A', and S11A), females for $w^{1118}/w^{1118}$; +/+; +/+ or $w^*$, $Ptp10D^1/w^*$, $Ptp10D^1$; +/+; +/+ were mated with males for $Ptp10D^1/Y$; +/+; +/+ and reared on 25 degrees. Selected F1 females for $w^*$, $Ptp10D^1/w^{1118}$; +/+; +/+ or $w^*$, $Ptp10D^1/Ptp10D^1$; +/+; +/+ were matured for 2 days on 25 degrees, and *Canton-S* males were matured for 5 days on 25 degrees. At this time, males and females were moved to vials individually with $CO_2$ anaesthesia 1 day before mating. After 1 day interval from $CO_2$ anaesthesia, single female and male were moved to new vials (Φ20 mm) with standard fly food for mating, and then measurement of egg laying was started on 25 degrees. For counting numbers of egg production, male and female were moved to new vial every day, and numbers of laid eggs and eggshells were counted manually using Leica M80 stereo microscope (Leica microsystems).

For the analysis of (Figs 6B, 6B', and S11B), females for $w^*/w^{1118}$; UAS-Dcr-2/tub-Gal80$^{ts}$; UAS-LUC.IR$^{TRiP.JF01355}$/bab1-Gal4 or $w^*/w^{1118}$; UAS-Dcr-2/tub-Gal80$^{ts}$; UAS-Ptp10D.IR$^{GD115}$/bab1-Gal4 in which were UAS-targeted sequences were induced from 1 day before wandering stage to 1 day after puparium formation with bab1-Gal4/TARGET were prepared as described in (see Materials and Method, Temporal expression of genes with TARGET system). Selected female with *LUC* RNAi (control) or *Ptp10D* RNAi were matured for 4 days on 18 degrees, and *Canton-S* males were matured for 5 days on 25 degrees and mated females and males were reared on 18 degrees for measuring the egg production. We extruded the samples from the data acquisition in cases that females did not lay egg in 2 days after mating and flies were died.

## Statistics

Raw data were tallied and processed with Microsoft 365 Excel (Microsoft). Graphs were designed with KaleidaGraph version 4.1.0 (Synergy Software) and Adobe Illustrator CC (Adobe). Wilcoxon rank-sum test was performed with KaleidaGraph for comparing medians of two sample distributions. Kruskal-Wallis rank sum test and Mann-Whitney U test with Bonferroni correction were performed with Easy R (EZR) version 1.54 [57] as one-way analysis of variance and multiple pairwise test, respectively, for comparing medians of more than two sample distributions. Fisher's exact test was performed with EZR for comparing the proportions of two samples. Pairwise comparison using Fisher's exact test with Bonferroni correction was performed with EZR for comparing the proportions of three samples.

## Supporting information

**S1 Movie. Germarium of wild-type (Fig 1B).**
(MP4)

**S2 Movie. Germarium of *Ptp10D[1]* (Fig 1C).**
(MP4)

**S1 Fig. *Gal4*-expression patterns of *Gal4*-drivers used in the experiments.**
(PDF)

**S2 Fig. Germarium phenotype by another *Ptp10D* RNAi line.**
(PDF)

**S3 Fig. Excessive GSCs contact with the abnormal cap cell-cluster in *Ptp10D[1]*.**
(PDF)

**S4 Fig. Morphologies of cap cells and surrounding muscles in *Ptp10D[1]*.**
(PDF)

**S5 Fig. Validation of antibodies and RNAi lines.**
(PDF)

**S6 Fig. Blocking apoptosis causes increased number of GSCs.**
(PDF)

**S7 Fig. Apoptosis in apical cells during larval gonadogenesis.**
(PDF)

**S8 Fig. Overexpression of apoptosis inhibitors suppress apoptosis in apical cells.**
(PDF)

**S9 Fig. Validation of *puc*-Stinger, a high-sensitive reporter of JNK signaling activity.**
(PDF)

**S10 Fig. Apoptosis and JNK signaling activity in gonadal apical cells.**
(PDF)

**S11 Fig. Loss of Ptp10D diminished the egg production.**
(PDF)

**S12 Fig. *Ptp10D[1]* does not exhibit increased apoptosis in oogenesis.**
(PDF)

**S1 File. Genotypes of flies used in the experiments.**
(DOCX)

**S2 File. Sequence map of *P-puc-Stinger*.**
(DOCX)

**S1 Dataset. Raw dataset.**
(XLSX)

## Acknowledgments

We thank S. Ohsawa for helpful discussions, M. Matsuoka, K. Gomi, and M. Koijima for technical support, Kai Zinn, Tian Xu, Romain Levayer, the Bloomington Drosophila Stock Center (BDSC), the Fly Stocks of National Institute of Genetics (NIG-FLY), KYOTO Drosophila Stock Center at Kyoto Institute of Technology (DGGR), and the Vienna Drosophila Resource Center (VDRC) for fly stocks, and the Developmental Studies Hybridoma Bank at The University of Iowa (DSHB) for antibodies.

## Author Contributions

**Conceptualization:** Kiichiro Taniguchi, Tatsushi Igaki.

**Data curation:** Kiichiro Taniguchi.

**Formal analysis:** Kiichiro Taniguchi.

**Funding acquisition:** Kiichiro Taniguchi, Tatsushi Igaki.

**Investigation:** Kiichiro Taniguchi.

**Project administration:** Kiichiro Taniguchi, Tatsushi Igaki.

**Supervision:** Tatsushi Igaki.

**Validation:** Kiichiro Taniguchi, Tatsushi Igaki.

**Visualization:** Kiichiro Taniguchi.

**Writing – original draft:** Kiichiro Taniguchi, Tatsushi Igaki.

**Writing – review & editing:** Kiichiro Taniguchi, Tatsushi Igaki.

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
