## [Decision Letter · Decision Letter 0]

1 Dec 2022

Dear Tatsushi,

Thank you very much for submitting your Research Article entitled 'Sas-Ptp10D shapes germ-line stem cell niche by facilitating JNK-mediated apoptosis' to PLOS Genetics.

The manuscript was fully evaluated at the editorial level and by independent peer reviewers. The reviewers appreciated the attention to an important problem, but raised some substantial concerns about the current manuscript. Based on the reviews, we will not be able to accept this version of the manuscript, but we would be willing to review a much-revised version. We cannot, of course, promise publication at that time.

If you decide to revise the manuscript for further consideration at PLOS Genetics, please aim to resubmit within the next 60 days, unless it will take extra time to address the concerns of the reviewers, in which case we would appreciate an expected resubmission date by email to plosgenetics@plos.org.

We are sorry that we cannot be more positive about your manuscript at this stage. Please do not hesitate to contact us if you have any concerns or questions.

Yours sincerely,

Norbert Perrimon

Academic Editor

PLOS Genetics

Gregory P. Copenhaver

Editor-in-Chief

PLOS Genetics

Reviewer's Responses to Questions

**Comments to the Authors:**

Reviewer #1: In this study, the authors investigate the role of a signaling cascade mediated by the transmembrane protein, Sas, and its receptor, Ptp10D, in the development of the Drosophila ovarian germline stem cell niche. The study begins with the observation that the adult ovaries of Ptp10D[1] mutant homozygotes have more germline stem cells than wildtype. Using standard genetic tools in the field to restrict RNAi expression in time and space, they demonstrate that Ptp10D and sas are required in the surrounding somatic cells specifically during a midlarval stage. Subsequent analysis provides evidence that the effect on adult stem cell number is caused by a role for this pathway in determining niche shape by promoting JNK-mediated apoptosis of apical cells during development via inhibition of egfr. The study is well-designed and, in general, the data provide clear support for the model. However, several issues would be important to address prior to publication.

1. The RNAi phenotypes in Fig. 1E-F are subtle, especially compared to the phenotype in the Ptpn10D null (Fig. 1D) and it is not clear if this is because the RNAi knockdown is incomplete or another reason. To test the efficiency of knockdown, the authors should use the anti-Ptp10D antibody and Sas::SGFP line in combination with the RNAi.

2. Likewise, the effect of Ptp10D RNAi knockdown on the rate of egg production is modest. Expression of Sas RNAi with bab1 had a stronger effect on GSC number--one that looks more similar to the effect in Ptp10D[1] mutants. What is the rate of egg laying in these mutants? Quantifying egg laying in additional mutants would help strengthen the correlation between increased GSC number and decreased egg laying.

3. As a minor comment (optional to address), displaying the rate of egg laying per ovariole rather than per female has advantages but is also unconventional. Can a separate graph or table be provided that shows the number of eggs laid per female so that these data can be more directly compared to egg laying assays in other papers?

4. The effect on apoptosis suggests that there should be more cap cells in Sas and Ptp10D mutants. Is this the case?

5. An important test of the proposed model would be to determine whether the miniCic::mCherry signal is altered in the Ptp10D and/or Sas mutants.

Reviewer #2: In Drosophila oogenesis, a cluster of cap cells form the niche for 2-3 germline stem cells (GSCs). In analyzing a viable mutant of Ptp10D, the authors discovered additional GSCs, an abnormal cap cell cluster structure and reduced fecundity. The phenotype was suppressed by heterozygosity for EGFR, suggesting that Ptp10D controls cap morphology via a similar pathway to cell competition where EGFR inhibits JNK-mediated apoptosis.

Using Gal4 drivers, the authors attempt to determine the cell specific requirement for Ptp10D and its ligand Sas. While they are able to show that the genes are required somatically, determining the exact cell type is a bit complicated due to the complex patterns of the Gal4 drivers. They conclude that the genes are required in the apical cells, since tj-GAL4 does not recapitulate the phenotype with UAS-Ptp10D RNAi and tj is not expressed in the apical cells, whereas the bab-1 and c587 drivers are. The bab-1-Gal4 shows the strongest effect, and the strongest effect was observed by expressing RNAi one day before the wandering larval stage.

They find that EGFR signaling via a Cic reporter is enriched in the distal apical cells adjacent to the terminal filaments. However, this does not seem to make sense in light of their proposed model. In their model (Figure 7), active Ptp10D should reduce EGFR signaling. This needs to be explained further.

They go on to examine apoptosis, and find a modest effect on cap cell structure using RHGmir and p35. They examine JNK signaling via puc overexpression and signaling and see an effect, suggesting that JNK regulates apoptosis to affect cap morphology.

This is an interesting manuscript pointing to a role for using the cell competition mechanism in tissue morphology. While the effects on oogenesis are clear, there are still a number of missing links that need further clarification. These are itemized below.

Major issues:

1. The cellular basis for the abnormal shape of the cap cell cluster is not described. Are there more cells? Are the cells a different shape? Or is anchorage to other cells disrupted? Is there a problem with the sheath? Some insight along these lines would be very helpful.

2. What is the relationship between the apical cells and the future cap cells? How do the authors propose that the increase/survival of apical cells leads to the abnormal cap cell cluster?

3. The conclusion that it is the apical cells that are relevant is not entirely clear given the complexity of the GAL4 patterns. For example, why is the RNAi effect with bab-1 stronger than c587 given that c587 is expressed more strongly in the apical cells adjacent to the terminal filaments? Images in S1 Fig are of wandering larval gonads. Since the RNAi effect was most pronounced in “before wandering” larvae (Fig 1F), images of larval gonads at those stages should be provided. These may provide additional insight into which specific cells are important.

4. In Figure 2, it looks like most of the expression of Ptp10D is in the terminal filament cells, not apical cells as described. This needs to be explained. It is also intriguing that the Sas expression looks adjacent to the Ptp10D staining perhaps in the most basal of the apical cells.

5. Antibody staining on the Ptp10D mutant and knockdown larval gonads should be provided to demonstrate the specificity of the antibody.

6. In the apoptosis experiments, the effect in Figure 4C is quite modest, but the effect in D-G looks more convincing. It would be interesting to stain the p35 expressing flies for cDcp-1 or tunel as it is possible that p35 is not blocking the cell death effectively. In addition, blocking apoptosis with dronc RNAi or Diap1 might be more effective than the RHGmir and p35.

Minor points

1. Fig 1 B,C would be helpful to show 3D image (movie) to make it easier to see all of the GSCs.

2. They should provide numbers of GSCs in the apoptosis experiments (Figure 4) given how modest the effect was on cap shape.

3. They should look at the EGFR reporter in the Ptp10D mutant background to confirm that it directly regulates EGFR in this context.

4. The puc overlap with Dcp-1 is not convincing in Figure S6.

5. Discussion – the ratio of germ cells to follicle cells might cause more disruptions to oogenesis. This would be seen as increased apoptosis of germcell clusters in the germarium. Were any defects seen?

**Have all data underlying the figures and results presented in the manuscript been provided?**

Reviewer #1: **No: **I might be missing something, but I have not seen the raw data for this study. The review invitation included a link to view the manuscript, but that just prompted a download of a PDF file that contained the manuscript and figures; no raw data was included in that PDF.

Reviewer #2: Yes

PLOS authors have the option to publish the peer review history of their article (what does this mean?). If published, this will include your full peer review and any attached files.

Reviewer #1: No

Reviewer #2: No

---

## [Decision Letter · Decision Letter 1]

28 Feb 2023

Dear Tatsushi,

We are pleased to inform you that your manuscript entitled "Sas-Ptp10D shapes germ-line stem cell niche by facilitating JNK-mediated apoptosis" has been editorially accepted for publication in PLOS Genetics. Congratulations!

Yours sincerely,

Norbert Perrimon

Academic Editor

PLOS Genetics

Gregory P. Copenhaver

Editor-in-Chief

PLOS Genetics

Comments from the reviewers (if applicable):

Reviewer's Responses to Questions

**Comments to the Authors:**

Reviewer #1: With the revisions in this manuscript, the authors have fully addressed my concerns and I now support publication. The findings presented here will be an important contribution to the field.

Reviewer #2: The authors have provided a thorough response to the previous reviews and have addressed my concerns.

**Have all data underlying the figures and results presented in the manuscript been provided?**

Reviewer #1: Yes

Reviewer #2: Yes

PLOS authors have the option to publish the peer review history of their article (what does this mean?). If published, this will include your full peer review and any attached files.

Reviewer #1: No

Reviewer #2: No

**Data Deposition**

http://datadryad.org/submit?journalID=pgenetics&manu=PGENETICS-D-22-01271R1

**Press Queries**

---

## [Editor Report · Acceptance letter]

22 Mar 2023

PGENETICS-D-22-01271R1 

Sas-Ptp10D shapes germ-line stem cell niche by facilitating JNK-mediated apoptosis 

Dear Dr Igaki, 

We are pleased to inform you that your manuscript entitled "Sas-Ptp10D shapes germ-line stem cell niche by facilitating JNK-mediated apoptosis" has been formally accepted for publication in PLOS Genetics! Your manuscript is now with our production department and you will be notified of the publication date in due course.

With kind regards,

Anita Estes

PLOS Genetics

On behalf of:
